# Unraveling Downstream Gender Bias from Large Language Models: A Study on AI Educational Writing Assistance

**Thiemo Wambsganss**[*] [1]**, Xiaotian Su**[*] [2]**,**
**Vinitra Swamy** [2]**, Seyed Parsa Neshaei** [2]**, Roman Rietsche** [1]**, Tanja Käser** [2]

[1] Bern University of Applied Sciences

{thiemo.wambsganss, roman.rietsche}@bfh.ch

[2] EPFL, Lausanne, Switzerland

{xiaotian.su, vinitra.swamy, seyed.neshaei, tanja.kaeser}@epfl.ch

## Abstract

Large Language Models (LLMs) are increasingly utilized in educational tasks such as providing writing suggestions to students. Despite their potential, LLMs are known to harbor inherent biases which may negatively impact learners. Previous studies have investigated bias in models and data representations separately, neglecting the potential impact of LLM bias on human writing. In this paper, we investigate how bias transfers through an AI writing support pipeline. We conduct a large-scale user study with 231 students writing business case peer reviews in German. Students are divided into five groups with different levels of writing support: one classroom group with feature-based suggestions and four groups recruited from Prolific – a control group with no assistance, two groups with suggestions from fine-tuned GPT-2 and GPT-3 models, and one group with suggestions from pre-trained GPT-3.5. Using GenBit gender bias analysis, Word Embedding Association Tests (WEAT), and Sentence Embedding Association Test (SEAT) we evaluate the gender bias at various stages of the pipeline: in model embeddings, in suggestions generated by the models, and in reviews written by students. Our results demonstrate that there is no significant difference in gender bias between the resulting peer reviews of groups with and without LLM suggestions. Our research is therefore optimistic about the use of AI writing support in the classroom, showcasing a context where bias in LLMs does not transfer to students' responses [1].

## 1 Introduction

Large Language Models (LLMs) have proven useful for improving the adaptivity and individualization of educational technology (Jones and Steinhardt, 2022; Xu et al., 2021). Researchers and practitioners have been developing a plethora of educational writing tools based on Natural Language Processing (NLP), for example for writing suggestions, (Wambsganss et al., 2022a; Lauscher et al., 2018) conversational interaction (Ruan et al., 2019; Schmitt et al., 2021), or to support writing skills and learning at scale. Nevertheless, the increasing use of LLMs for personalized support (especially in education) bears issues of critical concern. Research has found that these models can carry and propagate significant bias (Bolukbasi et al., 2016; Sun et al., 2019). LLMs, at any stage in their development pipeline, can harbor and propagate biases (e.g., gender, racial, or conceptual) and thus reflect the data they are trained on (Baker and Hawn, 2021; Hutchinson and Mitchell, 2019; Sun et al., 2019). Such biases, specifically gender bias, in LLMs, can reinforce harmful stereotypes in automated writing support applications, for example in automated essay scoring (Östling et al., 2013; Yannakoudakis et al., 2011) or individual writing support (Wambsganss et al., 2022b), and thus can inadvertently influence students' writing styles and perspectives (Su et al., 2023; Lee et al., 2022a).

Previous research has explored the existence of bias in different language models and their representations, focusing on the English language (Baker and Hawn, 2021). However, a growing body of literature advocates for the necessity of conducting comprehensive bias analyses, i.e. analyzing the impact of language models when embedded in human educational tasks (Lee et al., 2022b; Baker and Hawn, 2021; Blodgett et al., 2020). Especially for gender stereotypes, recent studies such as Andersson et al. (2021) and Cheng et al. (2022) have demonstrated the effects of such stereotypes on CV screening and child welfare programs, respectively. Nevertheless, detailed examinations of the effects of LLM-based writing suggestions on students and their use of gender stereotypes have been rather sparsely investigated, especially in other lan-

---

[*]These authors contributed equally to this work

[1]Our code and data available at: https://github.com/epfl-ml4ed/unraveling-llm-bias

guages than English (Baker and Hawn, 2021; Lee and Kizilcec, 2020). Given the expanding use of these models in educational settings and for writing assistance in general, (e.g., (Lee et al., 2022a; Chang et al., 2023), addressing this literature gap is of utmost importance. Understanding the nuanced ways in which biases in LLMs can seep into educational tasks can help researchers create safer and more effective learning environments that promote equitable outcomes.

In this paper, we analyze how bias transfers through an AI writing support pipeline and we investigate whether bias in LLMs translates into bias in human writing. We use an educational context, namely German peer reviews collected from 231 students divided into five groups: one group of 52 students in a University classroom setting receiving feedback from a traditional feature-based recommender system (G0) and four groups (179 students in total) recruited through Prolific. Groups G1-G4 include a control group receiving no writing support (G1), two groups with suggestions from GPT-2 (G2) and GPT-3 models (G3) fine-tuned on an extended version of the non-biased corpus of Wambsganss et al. (2022b) containing $11,925$ peer reviews in German, and one group with suggestions from pre-trained GPT-3.5 (G4).

We apply the GenBit gender bias test Bordia and Bowman (2019), the German adaptation Kurpicz-Briki (2020) of the Word Embedding Association Tests (WEAT) (Caliskan et al., 2017), and Sentence Embedding Association Test (SEAT) (May et al., 2019) translated to German to the collected peer reviews as well as to the suggestions provided by the different LLMs and the embeddings of fine-tuned GPT-2 model. With our analyses, we aim to answer the following two research questions:

1. In a real-world peer review writing exercise with AI writing support, does LLM bias transfer to student writing (RQ1)?

2. How does bias transfer across the different stages (i.e., model embeddings, model suggestions, student output) of the AI writing support pipeline (RQ2)?

Our results reveal a promising trend: groups receiving suggestions from LLMs (G2-G4) exhibit **no significant measurable difference in gender bias** in their writing compared to the control group without writing support (G1) and the in-classroom students receiving recommender-based feedback

(G0). Furthermore, none of the GenBit gender bias, WEAT tests and SEAT tests reveal biases in the provided suggestions from any of the LLMs, despite that the analysis of GPT-2 embeddings detects significant gender bias for the GPT-2 model. Our results therefore demonstrate that LLM-based writing support can be positively used for specific educational scenarios without bias.

## 2 Related Work

### 2.1 Bias in Educational Writings

Research has analyzed bias in educational technology since around the 1960s and many parts of today's research on algorithmic bias and fairness have been anticipated (Baker and Hawn, 2021). To effectively probe bias, it is imperative to establish our viewpoint, as the term "bias" is multifaceted and defined differently across various research works (see literature reviews such as Hutchinson and Mitchell (2019); Baker and Hawn (2021)). In our study, we adopt the view of algorithmic bias as "situations where model performance is substantially better or worse across mutually exclusive groups" (Baker and Hawn, 2021, p. 4). LLMs, such as GPT-2, GPT-3, or BERT, have been increasingly utilized for educational writing assistance (Mirowski et al., 2023; Gero et al., 2022). These models, trained on extensive and diverse data, have proven instrumental in predicting subsequent text, thereby producing coherent responses (Lee et al., 2022a,b). Research has studied the risk that they might reflect by investigating the biases inherent in the training data and the models (Kurpicz-Briki, 2020). We aim to scrutinize the impact of bias from LLMs on student writings, thereby focusing on the lower end of the NLP pipeline, investigating the impact of writing suggestions on educational downstream tasks as Lee et al. (2022b) suggested.

### 2.2 NLP Bias Analysis

Research within the field of computational linguistics and NLP has progressively delved into bias present in language systems. This includes investigations into bias in areas such as embedding spaces (Caliskan et al. (2017); Bolukbasi et al. (2016)), language modeling (Lu et al., 2018), co-reference resolution (Rudinger et al., 2017), machine translation (Stanovsky et al., 2019), and sentiment analysis (Kiritchenko and Mohammad, 2018). Sun et al. (2019) have investigated strategies to mitigate gender bias across various NLP tasks. A variety

of methods have been proposed for the detection of gender bias in text. These range from multi-dimensional classifications of gender bias (Dinan et al., 2020) to the exploration of the frequency of gender bias metrics using word embeddings (Valentini et al., 2022). Some studies have even analyzed human gender stereotypes via word association tests (Du et al., 2019). Instruments such as the Word Embedding Association Test (WEAT) (Caliskan et al., 2017) are commonly utilized for bias identification, enabling the quantification of biases within word embeddings by assessing the strength of correlations between target words and attribute words (Du et al., 2019). WEAT is composed of different tests that aim to reveal racial, conceptual, and gender bias. WEAT tests 6, 7, and 8 have been also used to investigate gender bias in German texts (Kurpicz-Briki, 2020). Furthermore, Bordia and Bowman (2019) proposed a test to detect gender bias in word-level language models and suggested a bias score that has been widely used in the NLP community (e.g., Sengupta et al. (2021)). For sentence-level bias analysis, there is the Sentence Embedding Association Test (SEAT) (May et al., 2019) which extends WEAT to measure bias in sentence encoders.

## 2.3 Studies on Bias in Writing Assistance

Nevertheless, studies that investigate biases in downstream applications and particularly the influence of LLMs on human writing are rather rare. While Lee et al. (2022b) have proposed a framework for investigating the impact of LLMs on human writing, they have only motivated and not investigated the impact of toxicity and bias on human texts after receiving writing assistance. Furthermore, studies in the educational domain involving real students especially outside of North America remain limited (Baker and Hawn, 2021; Sun et al., 2019). Our work centers on the impact of writing suggestions provided by LLMs on students' gender stereotypes. Our goal extends past work on revealing bias in the educational NLP pipeline (Wambsganss et al., 2022b) and on human evaluation (Lee et al., 2022b) by shedding light on the impact of potentially biased LLMs on students. We do so by investigating the case of student peer reviews in the German language since this is a domain-independent and increasingly embedded educational context fostering writing competencies in large-scale learning scenarios. With this, we aim

to contribute towards shaping a future where NLP researchers and practitioners are aware of biases of downstream educational models and hence strive to minimize the potential harm by those models when applied in sensitive contexts like education, potentially involving sensitive user groups (such as under-aged students).

## 3 Methodology

To investigate the impact of LLMs on gender biases in human writing within an educational context, we performed a bias analysis in three steps (see Figure 1). In a first step, we trained a feature-based recommender system and fine-tuned GPT-2 and GPT-3 models on a corpus of $11,925$ student-written business peer reviews in German to be able to provide automated suggestions in a peer review exercise. In a second step, we conducted a large-scale user study with 231 students in a classroom and an online context, providing students with different levels of writing support (feature-based recommender, no support, suggestions by fine-tuned GPT-2 or GPT-3, suggestions by pre-trained GPT-3.5) in a peer review writing exercise in German. In the third and final step, we used the GenBit Gender Bias Score (Bordia and Bowman, 2019), WEAT, and SEAT tests to analyze the gender bias of the resulting reviews, the model suggestions, as well as the GPT-2 embeddings.

## 3.1 Model Development

In the FUSM group (G0), students received automated advice based on the Feature Utility Saturation Model (FUSM). This model is trained on a corpus of $9,000$ student-written peer reviews in German. The recommender system predicts the utility of specific types of feedback in improving the review quality (Bauman et al., 2020). A personalized subset of 3 out of 15 features, such as adding suggestions, decreasing sentence length, and writing more directive words, is provided upon clicking the *get advice* button. The features have been derived based on existing literature on peer feedback and a dictionary with keywords was developed to operationalize the features. We followed Bauman et al. (2020) for the implementation of our model.

To provide students with adaptive writing suggestions, we fine-tuned GPT-2 and GPT-3 models (used by groups G2 and G3) and embedded GPT-3.5 in the peer review tool (used by group G4). The two models for G2 and G3 were fine-tuned on

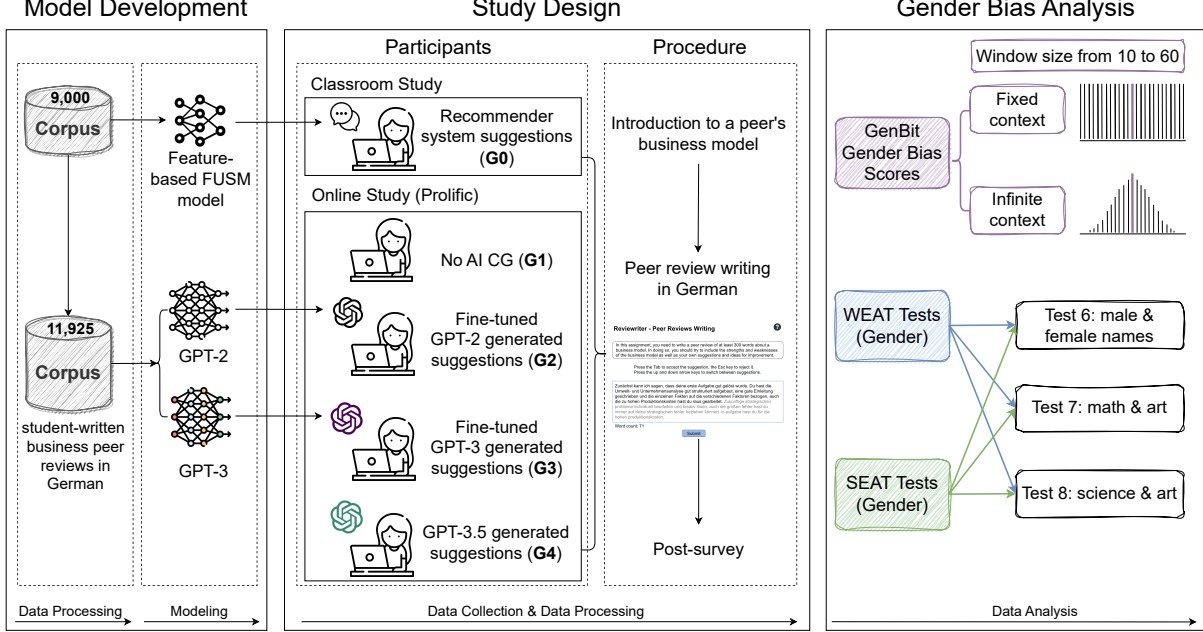

Figure 1: Overview of our pipeline. We first prepared machine learning models as well as LLMs to provide writing suggestions for learners, see Section 3.1 for more details. We then conducted a user study with a peer review writing exercise. Section 3.2 presents the whole study design including the procedure and all five groups finished the exercise with different writing assistance tools. We provided details on data collection as well as data processing. Finally, we analyzed gender bias using the GenBit Bias Score, WEAT and SEAT tests (see Section 3.3).

an extended version of the non-biased corpus of peer reviews in German reported by Wambsganss et al. (2022b). The data was collected over four years and includes 11,925 reviews from 610 unique reviewers and 607 reviewees. Students wrote approximately nine peer reviews per course with an average length of 220 words. This extensive corpus served as a solid foundation for fine-tuning models and preparing them to provide writing suggestions to students. We started data processing by expanding abbreviations, removing HTML tags, irrelevant information like PDF file names and specific information like URLs, keywords (revealing the identity of students), and questions asked to write reviews which some students copied to their review text. Then, we shuffled and divided the cleaned data into train and test datasets with proportions of 0.8 and 0.2 for fine-tuning and evaluating the language models.

### 3.2 Study Design

**Participants.** Participants of the user study were 231 students who were split into five distinct groups (G0 - G4), controlling for the sensitive variables of education level, language, age, and gender. While there might be (small) observable differences in the sample means or medians for sensitive variables,

additional statistical analysis confirms that there are no significant differences between groups and that the randomization has worked correctly. For the age attribute, a Shapiro-Wilk test indicates that the data is not normally distributed and we have therefore conducted a non-parametric Kruskal-Wallis test, which confirms that there are no significant differences between groups G1-G4 in terms of age ($H = 2.24$, $p = .52$). Furthermore, a Chi-Square test confirms that there are also no significant differences between the four Prolific groups in terms of gender ($X2 = 0.0149$, $p = .99$) or education level ($X2 = 6.65$, $p = .35$). Each group was provided with writing suggestions from a unique model: G1 received suggestions from a feature-based recommender system, for example, on text length or sentiments, G2 received suggestions from a fine-tuned GPT-2 model, G3 from a fine-tuned GPT-3 model, and G4 from pre-trained GPT-3.5. Table 1 provides an overview of each group and the demographics of the participants.

**Procedure.** In order to control for the content aspect (as the content of the provided business model could inadvertently have an influence on the expected bias), we present each participant with exactly the same predefined business model.

| Context | Group | # | Gender | | Age | | Highest Education | |
|---------|-------|---|--------|--------|------|------|-----------|------------|
| | | | Male | Female | Mean | Std. | Below BSc | BSc and above |
| *Classroom* | FUSM (G0) | 52 | 62.0% | 38.0% | 25.7 | 1.9 | 0.0% | 100% |
| *Online* | None (G1) | 40 | 50.0% | 50.0% | 30.0 | 8.3 | 30.0% | 70.0% |
| | GPT-2 (G2) | 50 | 50.0% | 50.0% | 28.3 | 8.5 | 22.0% | 78.0% |
| | GPT-3 (G3) | 44 | 50.0% | 50.0% | 29.9 | 12.6 | 34.1% | 65.9% |
| | GPT-3.5 (G4) | 45 | 54.3% | 45.7% | 30.9 | 12.3 | 39.1% | 60.9% |

Table 1: Overview of the data sample and participant demographics of our user study across five groups.

Students in the FUSM group (G0) received writing suggestions through a dashboard next to the text input in the form of syntactical and semantic advice. We collected data in a lecture at a Western European university where $52$ students were writing up to three reviews on a peer's business model and they went through three peer feedback rounds.

The data of the three LLMs groups (G2-G4), as well as the control group (G1), was collected through the online platform Prolific[2] in order to not cause potential harm to real-world students. The task in the Prolific study involved the participants writing a review on a pre-defined business model. Specifically, participants were asked to elaborate on strengths, weaknesses, and suggestions for improvement of the given business model. We instructed people not to use search engines and spend a minimum of 15 minutes on the task. A countdown indicated the remaining time. Students received different forms of writing support on that task. Specifically, we followed the interface of the human-centric educational tool of Su et al. (2023) (see Figure 4 in the appendix) to ensure that students received beneficial writing aid with a satisfactory user experience. We presented users with next-sentence predictions, providing three suggestions at each point in time. Students could use the *Tab* key to accept a suggestion, the *Esc* key to reject a suggestion, and the *Up* or *Down* arrow keys to toggle through different suggestions. During the writing process, we collected the final writings, suggestions, as well as keystrokes of participants.

We cleaned the collected data by removing HTML tags, emojis, punctuation, abbreviations, digits, and stop words (excluding words in the gender lists [3]). The text was transformed to lower-

case. Because of the linguistic characteristic of the German language and its grammatical genders, special words like *Ihr*, *Ihrem*, *Ihren*, *Ihrer*, *Ihres*, and *Sie* have both gendered meanings ("she") and non-gendered meaning (e.g. "you"). We filtered them by checking their position in the sentence. If they related to the polite form (*Höflichkeitsform*), we removed them to not inflate the bias score in the later calculation. Otherwise, we kept them. Afterward, we did a human evaluation. Three German researchers checked the outputs manually to confirm the quality of the processed data.

### 3.3 Gender Bias Analysis

To investigate the bias in the resulting peer reviews, the model suggestions, and the model embedding, we utilized three different gender bias tests.

#### 3.3.1 GenBit Gender Bias Score

The GenBit Gender Bias Score was introduced by Bordia and Bowman (2019) and is included in the Microsoft Responsible AI Toolkit (Sengupta et al., 2021). The idea of the method is to identify gender bias by measuring the association between pre-defined gendered words and other words in the corpus via co-occurrence statistics.

**Bias score definition.** To compute the bias score, we first estimate the probability of a word occurring in the context of gendered words within a text corpus. The probability $P(w|g)$ indicates how likely it is for a specific word, denoted as $w$, to appear near (within a context window $k$) gendered words, denoted as $g$:

$$P(w|g) = \frac{\text{count}_k(w, g)}{\sum_i \text{count}_k(w_i, g)}$$

where $w$ is any word in the corpus, excluding stop words and gendered words, $count_k(w, g)$ represents the count of occurrences where the gendered

[2]www.prolific.co
[3]https://github.com/epfl-ml4ed/
unraveling-llm-bias/tree/main/GenBit/gender%
20list

word $g$ appears in the context window $k$ of the target word $w$ and $\sum_i count_k(w_i, g)$ calculates the total count of occurrences where any word $w_i$ appears in the context window $k$ of gendered words $g$. Finally, $g$ is the set of gendered words that belong to either of the two categories: male or female. In other words, we count how many times the word $w$ and at least one gendered word from the set $g$ appear within a certain distance of each other. In terms of defining gender pairs, we adopted the same set of gender term pairs as Sengupta et al. (2021). Additionally, given the linguistic characteristic of the German language, we added gendered pronouns to better capture the grammatical gender. For pronouns with ambiguous meanings, we did a filtering process in the data processing stage under the instruction of three German researchers.

The bias score of a specific word $w$ is then defined as:

$$bias(w) = \log(\frac{P(w|m)}{P(w|f)})$$

This bias score is measured per word and review. A positive bias score implies that a word co-occurs more often with male words than female words.

**Windows size and context window.** For a context size *k*, there are *k/2* words before and *k/2* words after the target word *w* for which the bias score is being measured. Qualitatively, a smaller context window size has more focused information about the target word. On the other hand, a larger window size captures topicality (Levy and Goldberg, 2014). According to Bordia and Bowman (2019), the optimal window size for fixed context is $k = 20$, which assigns an equal weight of $5\%$ to the ten words before and the ten words after the target word. For an infinite context, weights diminish exponentially based on the distance between the target word $w$ and the gendered word $g$. This method emphasizes the fact that the nearest word has more information about the target word.

### 3.3.2 Word Embedding Association Test

To assess bias along the NLP pipeline suggested by Hovy and Prabhumoye (2021), we rely on the Word Embedding Association Test (WEAT) proposed by Caliskan et al. (2017). WEAT calculates the semantic similarity between two sets of target words (e.g., male vs. female names) and two sets of attribute words using word embeddings (e.g., career vs. family). Table 6 in the appendix indicates the

nine WEAT tests and their corresponding targets and attributes, which we translated into German. Our analysis focuses on the gender bias dimension of WEAT using tests 6, 7, and 8.

To quantitatively compare across WEAT analyses, we use the *effect size* as proposed by (Caliskan et al., 2017). Effect size is a normalized measure of the distance between the two distributions of associations and targets, calculated as follows:

$$\frac{\frac{1}{|X|} \sum_{x \in X} s(x, A, B) - \frac{1}{|Y|} \sum_{y \in Y} s(y, A, B)}{S_{w \in X \cup Y}(s(w, A, B))}$$

where $X$ and $Y$ are two sets of target words of equal size, $A$, $B$ are two sets of attribute words, $s(w, A, B)$ measures the association of embeddings of the target word $w$ with the attribute words, and $S$ denotes the standard deviation. We conduct WEAT analyses at two granularities as proposed by (Wambsganss et al., 2022b): in the raw text corpora (through co-occurrence and GloVE models) and in the fine-tuned language model.

### 3.3.3 Sentence Embedding Association Test

In addition to word-level metrics like GenBit and WEAT, we have sentence-level metrics, the SEAT test, for a more comprehensive analysis. This test was defined in May et al. (2019) and implemented in Meade et al. (2022). We used the implementation from Fairpy (Viswanath and Zhang, 2023), an open source Toolkit for measuring and mitigating biases in large pre-trained language models. By applying WEAT to the vector representation of a sentence, SEAT compares sets of sentences instead of sets of words. In the ideal case, the embedding representation of each word in the vocabulary is expected to be equidistant from the two attribute classes. Any deviation suggests bias in one direction. The greater the deviation, the greater the bias (Viswanath and Zhang, 2023). Same as WEAT tests, our analysis focuses on the gender bias dimension of SEAT using tests 6, 6b, 7, 7b, 8, 8b. However, to the best of our knowledge, there is no available German version of SEAT tests. We first translated the templates from English to German with the translation software, then the translations were revised by two native German speakers. To facilitate future work, we publicly provided the translated files [4].

---

[4] https://github.com/epfl-ml4ed/unraveling-llm-bias/tree/main/SEAT/translated%20de

## 4 Results

To evaluate whether bias transfers from AI assistants to students for a real-world writing support scenario (RQ1) and to investigate where in the pipeline the bias persists (RQ2), we applied the GenBit Bias score as well as the WEAT tests to the five different subsets from our user study: four from the Prolific user study (control group G1, GPT-2, GPT-3, and GPT 3.5 assisted reviews G2-G4) and one from the classroom study with recommender system suggestions (G0).

### 4.1 RQ1 - Does Bias Transfer?

| Group | # | GenBit Bias Score |
|---|---|---|
| FUSM (G0) | 310 | -0.024 ± 0.275 |
| Control (G1) | 40 | 0.065 ± 0.487 |
| GPT-2 (G2) | 50 | -0.099 ± 0.486 |
| GPT-3 (G3) | 44 | 0.115 ± 0.570 |
| GPT-3.5 (G4) | 45 | -0.058 ± 0.592 |

Table 2: Total number of reviews from each group (#) and statistical summary of the GenBit bias score for all groups. The large standard deviations indicate the variability of the range of bias scores. Traditional ML feedback (G0) has the lowest variability, while the LLM feedback (G2-G4) and the control group with no feedback (G1) have similar ranges of variability.

To answer our first research question, we computed the GenBit Bias Score for the peer reviews written by the students. The goal was to identify which biases in the models transfer to humans through collaboration. We experimented with both a fixed context for windows sizes ranging from 10 to 60 as well as an infinite context. Since there was no significant difference between the results from the fixed context and infinite context, and the optimal window size was determined to be 20 by previous work Bordia and Bowman (2019), in this section, we only present results for a fixed context with a window size of 20. The resulting GenBit bias scores are illustrated in Figure 2. The total number of reviews from each group and statistical summary of bias scores for all groups are presented in Table 2. We observe that all five groups exhibit bias scores close to 0. Additional results are provided in the appendix in Section B.

To analyze differences between the bias scores of the four Prolific groups (G1-G4), we first aggregated the mean bias scores of each review from these groups. The results of a Shapiro-Wilk test

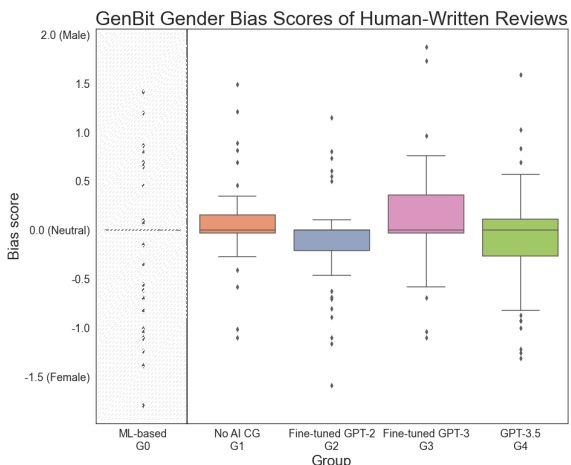

Figure 2: GenBit gender bias score of human-written reviews for a fixed context with a window size of 20.

(Shapiro and Wilk, 1965) then showed that our data was not normally distributed. Additionally, a Levene test (Levene, 1960) confirmed heteroscedasticity within the data. Therefore, we chose the non-parametric Mann-Whitney U test (MWU) to analyze the difference in mean bias scores between any pairing of two groups and the Kruskal-Wallis test (Kruskal and Wallis, 1952) for the difference among all four groups.

Table 3 presents the results of statistical tests analyzing the mean bias scores of reviews written with and without suggestions from LLMs. We did not find any statistically significant difference between the bias scores of the four groups.

| Group | p-value MWU Test GPT-2 (G2) | p-value MWU Test GPT-3 (G3) | p-value MWU Test GPT-3.5 (G4) |
|---|---|---|---|
| Control (G1) | 0.170 | 0.619 | 0.551 |
| GPT-2 (G2) | - | 0.075 | 0.635 |
| GPT-3 (G3) | - | - | 0.269 |

Table 3: Mann-Whitney U (MWU) test of bias scores for reviews from multiple groups. Asterisks indicate statistical significance (***: p<.001; **: p<.01; *: p<.05). There is no statistically significant difference between the bias scores of the four groups.

### 4.2 RQ2 - Where is Bias Present?

To answer our second research question and investigate where bias is present in the pipeline, we used the GenBit Gender Score, WEAT tests, and SEAT tests to analyze the LLMs' suggestions and (where available) the model's raw embeddings. This in-

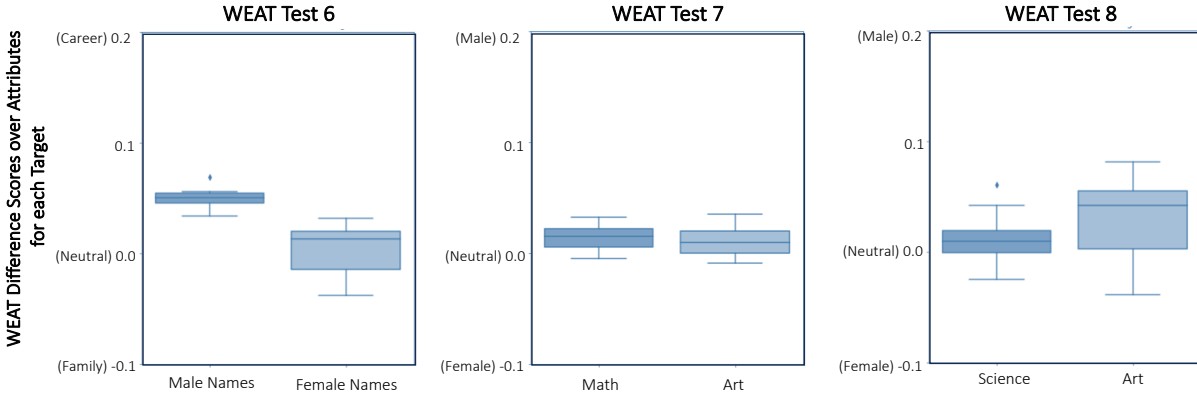

Figure 3: Comparing WEAT bias test scores for the fine-tuned GPT-2 model used in the human evaluation study.

depth analysis allowed us to dig deeper into the potential roots of bias within the suggestions made by the models and to further understand how these may have influenced the students' writings.

**Post-Hoc Analysis of Suggestions.** We measured the bias in the raw text corpora of model suggestions from fine-tuned GPT-2, GPT-3, and pre-trained GPT 3.5 with three methods: Gen-Bit bias scores, WEAT co-occurrence tests, and WEAT GloVE embedding tests. In each of these tests, we did not identify any significant bias in the suggestions. We pre-filtered suggestions that had more than 10 words to measure bias. On average, there were $13 - 15$ words in each suggestion across groups. For a fixed context with a window size of 10, the bias scores for fine-tuned GPT-2 and GPT-3 were 0.000092 and 0 respectively, and 0.000062 for GPT-3.5. In the GloVE embedding model trained on each text subset, we identified one instance of bias in GPT-2 suggestions in the classroom study for WEAT test 7 (examining female vs. male targets for math vs. art attributes), but this was considered insignificant. Overall, through the WEAT tests of co-occurence and embedding models, we determined that the raw text corpora across the study were unbiased.

**Post-Hoc Analysis of Model Embeddings.** To dive deeper into the bias behavior of a fine-tuned model used for writing assistance, we examined the gender bias of the fine-tuned GPT-2 model embeddings in both word-level and sentence-level. As shown in Figure 3 (left), we identified a notable bias in the fine-tuned model that was not present in the raw text corpora. In the WEAT Test 6, we examined attributes of Career vs. Family in relation to Male vs. Female names. We obtained an effect

score of $1.57$ (with effect scores ranging from $-2$ to $2$), indicating that the career attribute is much closer to Male than Female names.

WEAT Test 7 focuses on the target disciplines of Math and Art, while examining Male and Female as attribute lists. The difference scores in Figure 3 (middle) showed minimal disparity, showing a slight bias in favor of the male attribute for both Math and Art targets. The five most similar attribute words to the Math target list, translated from German to English, (young, son, brother, masculine, and sister) depicted a more extreme picture with 4 out of 5 male-oriented words, aligned with the 5 most similar attributes to Art (young, brother, son, man, and daughter). The effect size was $0.27$, showing that Math is slightly more aligned to the Male attribute than the Art target.

In WEAT Test 8, we examined the relationship between Science and Art targets and Male and Female attribute lists (see Figure 3 (right)). Again, we identified similarities in the top five attribute words with a strong male bias in the most related words. For the Science target, (uncle, son, father, daughter, and brother) were the five most similar attributes (four out of five male-oriented). For the Art target, the most similar attributes were (uncle, brother, son, father, and daughter) (again four out of five male-oriented). The effect size for this test was $-0.56$, indicating that Art is more related to male attributes than Science.

SEAT test results are summarized in Table 4, across tests 6, 6b, 7, 7b, 8, and 8b, which correspond to the analogous gender tests of WEAT, there is no significant difference found between the embeddings of the target sentences and attribute sentences (i.e. in SEAT 6, male and female names are both equally similar to career and family words),

| # | Tragets | Attributes | Effect size |
|---|---------|------------|-------------|
| 6 | Male vs. Female Names | Career vs. Family | 0.021 |
| 6b | Male vs. Female Terms | Career vs. Family | -0.074 |
| 7 | Math vs. Arts | Male vs. Female Terms | -0.705 |
| 7b | Math vs. Arts | Male vs. Female Names | -0.209 |
| 8 | Science vs. Arts | Male vs. Female Terms | -0.069 |
| 8b | Science vs. Arts | Male vs. Female Names | 0.078 |

Table 4: SEAT gender bias analysis for the fine-tuned GPT-2 model used in the human evaluation study.

computed using p-values and a hypothesis test as per the standard implementation. Each test also has an effect size, ranging from -2 to +2, with 0 representing a completely neutral effect between both target words and attribute words. Most effect sizes are within 0.1, so there are only minimal bias effects. The comparatively largest effect (SEAT test 7) is -0.705 from comparing Math and Arts in association with Male and Female terms, but this was still found to have insignificant bias. These results are in line with the WEAT analysis on the word level, which also found no significant gender bias in the model embeddings.

Interestingly, as demonstrated by our analyses, the gender biases revealed in GPT-2 embeddings did not translate into gender biases in suggestions.

## 5 Discussion and Conclusion

LLMs are increasingly used in educational settings, despite that they harbor inherent biases which may have a negative impact on learners. Our work analyzed how bias transfers through an AI writing support pipeline in an educational context and whether the bias in LLMs translates to bias in students' writings. To do so, we conducted a large-scale user study, providing students with different levels of writing support in a peer review writing exercise.

Our analysis of data collected from in-classroom and Prolific participants yielded positive results that provide optimism for the field of NLP and its application within educational contexts. The most notable finding was that students who received writing suggestions from LLMs exhibited the same degree of gender bias in their written text as students who received no suggestions. The group receiving suggestions from a recommender system with human interpretable features also showcased a similar amount of bias in student responses. Our results seem initially promising, showing that by measuring gender bias through multiple tests (Gen-Bit, WEAT, and SEAT) at each stage of the pipeline

(model embeddings, model suggestions, student output), LLMs do not inadvertently foster and perpetuate gender bias. One possible explanation for these results is that the biases present in the original training data and embeddings of the LLMs were not transferred to the model suggestions, as indicated by the lack of bias in model suggestions. Unbiased suggestions led to positive learning outcomes for students without an inadvertent bias transfer.

Our post-hoc deep dive into the bias dimensions of the suggestions and the GPT-2 embeddings revealed the inherited bias in the fine-tuned models. This analysis provided valuable insight into how biases can become ingrained in LLMs and also identified where in the pipeline the bias stops. It suggests that even when models are fine-tuned with the intention of reducing bias, their original training on vast amounts of data from the internet can leave an indelible imprint of bias. An important takeaway from our study is that the applied domain of education, although it is often a sensitive context, is moving towards integration with LLM assistants. Before we can be fully confident of model impacts on sensitive young minds for downstream tasks, we need to strive for not only more sophisticated bias detection and mitigation techniques but also more transparency in how these models are trained and fine-tuned. Also, future research is needed to better investigate the impact of potentially biased LLMs in different educational settings in addition to writing, for example, language learning (Xiao et al., 2023), STEM education (Lee and Perret, 2022), and legal education (Weber et al., 2023).

In conclusion, our study contributes towards a more nuanced understanding of how LLMs interact with bias for educational tasks, and produces the indication that although models contain bias, personalized downstream applications might not. We hope that our findings will stimulate further research, contributing to the UN's fourth sustainability goal of ensuring quality education for all.

## Limitations

While our study provides crucial insights into the impact of LLMs' potential bias on human writing, we acknowledge the need for further research. It would be insightful to extend our study to other generative language models, study populations (different student levels, different languages), and educational tasks to paint a more comprehensive picture of the complex dynamics at play.

The inability to have access to several of the advanced model embeddings, unfortunately, prohibited us from conducting a WEAT analysis of GPT-3 and 3.5; we make the informed assumption that bias exists in the embeddings as per previous work, but we cannot measure or compare it with our GPT 2 findings. Additionally, our educational task (a business case study regarding ski instruction) is neutral and does not easily lend itself to measuring gender bias, compared to other potential settings. It was selected because it is a real-world education example integrated currently in a Western European university. If the case study was regarding medicine, for example, the male and female words for the doctor would be different in German, and this could potentially lead to a different bias measurement. Other dimensions of bias evaluated in Liang et al. (2023); Lin and Ng (2023) (e.g., racial, cognitive) would also need to be studied before LLMs for writing support can be fully trusted in the classroom.

## Ethics Statement

We note that this research was conducted by a mixed team of authors with Western European, Asian, North-American, Middle Eastern, female, and male backgrounds. All student data was anonymized in the use of this study and the study was approved by the human research ethics commission of the university. While our results indicated that LLM-based AI writing support does not have adverse effects of bias transfer on student writing, our work is not a generalizable study over many student populations and many different exercise settings. We present a case study over several model types and a positive finding regarding classroom integration, but we still encourage caution and experimentation before integrating models in the classroom.

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

# A Interface of peer review writing during the Prolific study

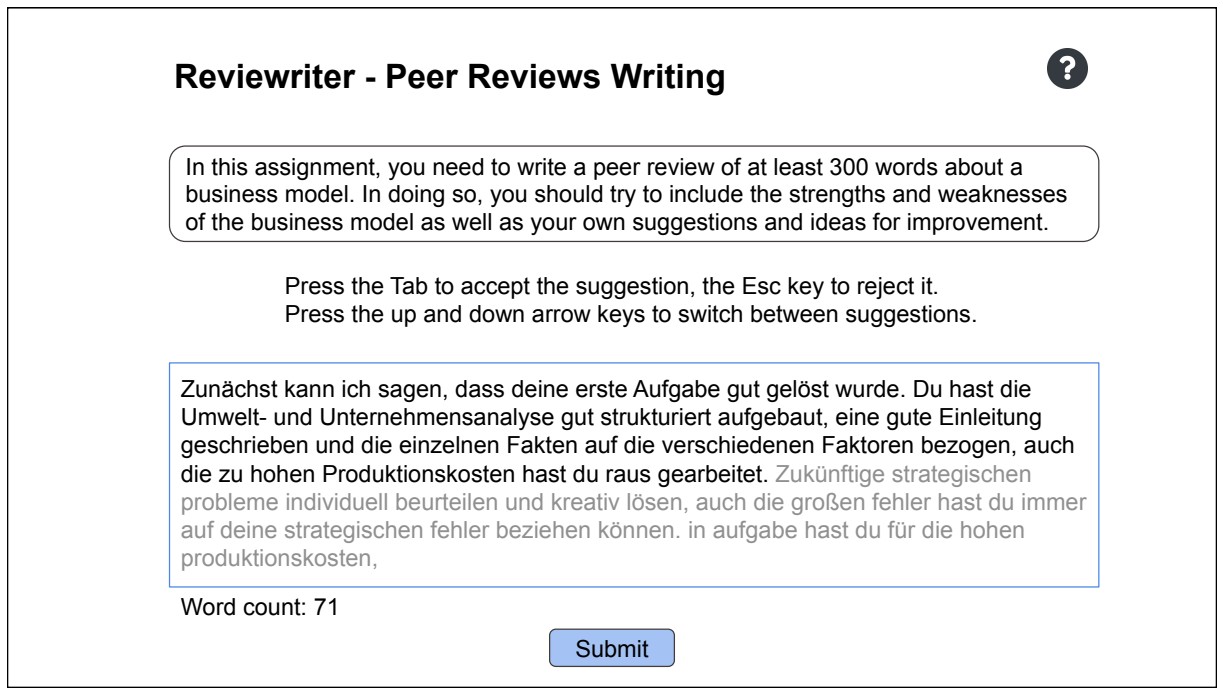

Figure 4: A screenshot of the interface to provide inline suggestions for peer review writing in the Prolific study. By clicking the question mark, people get detailed guidance on the peer review writing task and the usage of the tool. A simple text area supports all typical interactions, such as typing, selecting, editing, and deleting text, and caret movement via keys and mouse. In the input area, the sentences in black are the actual text, we display the AI-generated instruction in an inline format in gray. The model generates next-sentence predictions to give people a complete view of the idea. We provide three instructions each time, and people may use the *Tab* key to accept, the *Esc* key to reject, and the *Up* and *Down* arrow keys to toggle through different instructions). The total number of words is displayed below the text area to inform people of their writing progress.

# B Extra results of GenBit gender bias score with different window sizes

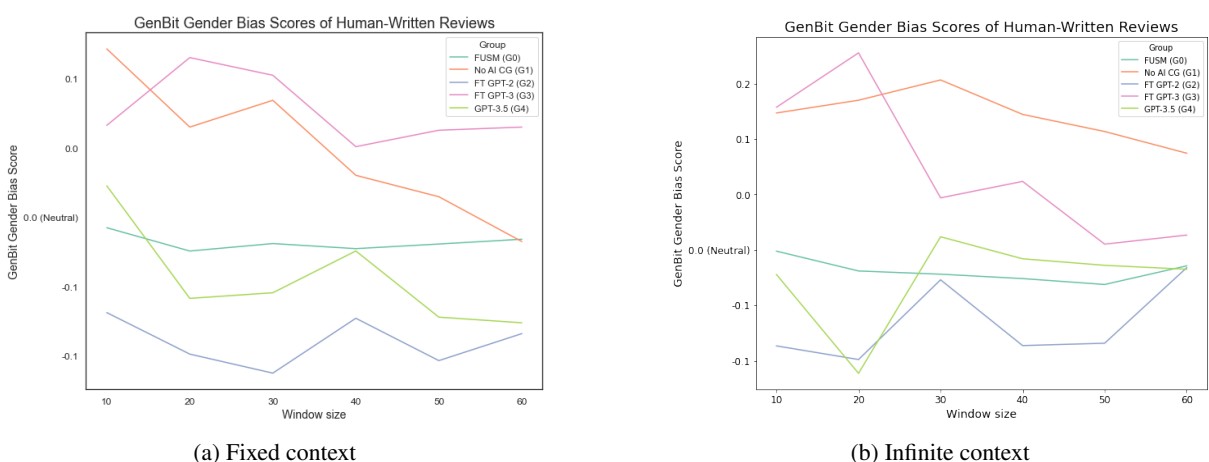

(a) Fixed context                    (b) Infinite context

Figure 5: Comparing Genbit gender bias scores with two different context types and varying windows sizes ranging from 10 to 60.

## C Statistics on gendered words

| Group | Avg. Num words | Avg. Num male words | Avg. Perc male words | Avg. Num female words | Avg. Perc female words |
|---|---|---|---|---|---|
| ML-based (G0) | 183 | 1.08 | 0.58% | 0.69 | 0.39% |
| No AI CG (G1) | 286 | 1.79 | 0.64% | 2.56 | 0.94% |
| FT GPT-2 (G2) | 299 | 2.16 | 0.71% | 2.18 | 0.71% |
| FT GPT-3 (G3) | 293 | 2.30 | 0.77% | 3.20 | 1.17% |
| GPT-3.5 (G4) | 291 | 2.71 | 0.93% | 2.98 | 1.03% |

Table 5: Statistics of total words and gendered words per review from each group after data processing.

## D WEAT analysis categorization form

| Bias | # | Targets | Attributes |
|---|---|---|---|
| | 1 | Flowers vs. Insects | Pleasant vs. Unpleasant |
| Conceptual | 2 | Instruments vs. Weapons | Pleasant vs. Unpleasant |
| | 9 | Mental vs. Physical Disease | Temporary vs. Permanent |
| | 3 | Native vs. Foreign Names | Pleasant vs. Unpleasant |
| Racial | 4 | Native vs. Foreign Names (v2) | Pleasant vs. Unpleasant |
| | 5 | Native vs. Foreign Names (v2) | Pleasant vs. Unpleasant (v2) |
| | 6 | Male vs. Female Names | Career vs. Family |
| Gender | 7 | Math vs. Arts | Male vs. Female Terms |
| | 8 | Science vs. Arts | Male vs. Female Terms |

Table 6: WEAT analysis categorization from Wambsganss et al. (2022b) in three dimensions (conceptual, racial, and gender). In this work, we use the gender dimension tests. WEAT compares the association between two different target word lists (i.e. Math vs. Arts) to attribute word lists (i.e. Male vs. Female terms). # indicates the original WEAT test number (Caliskan et al., 2017).