# OpenReview forum: "Unraveling Downstream Gender Bias from Large Language Models: A Study on AI Educational Writing Assistance"
_EMNLP/2023/Conference — EMNLP 2023 Findings_

### Official Review · Reviewer_YUR3 · 2023-08-02

**Soundness:** 3

**Excitement:**

3: Ambivalent: It has merits (e.g., it reports state-of-the-art results, the idea is nice), but there are key weaknesses (e.g., it describes incremental work), and it can significantly benefit from another round of revision. However, I won't object to accepting it if my co-reviewers champion it.

**Paper Topic And Main Contributions:**

The authors describe an analysis of the downstream impact of LLM usage on gender bias in the educational setting. This is studied across different groups of users that utilize different, or no, LLMs for writing suggestions in the peer reviewing process. The authors first analyze the final written reviews for biased language and find no significant differences across different groups of users. They then study where bias may stem from, in regards to the model embeddings and model suggestions. These results show gender bias in model embeddings that do not translate to bias in model suggestions.

**Questions For The Authors:**

Was each user shown the same pre-defined business model (ski instruction) in the Prolific studies or did this vary across participants?

While you mention that the context of the reviews are domain-independent, I wonder if the business models that are being reviewed are not related to topics that would easily exhibit gender bias to begin with? Do you have examples of the topics of these business models?

**Reasons To Accept:**

The authors highlight an import aspect of LLM usage in the educational setting for suggestions and rewrites. This is an emergent domain of study given the popularity of LLMs in general society in the past few months.

The authors use multiple gender bias metrics and models for their study.

The study evaluates multiple aspects of the NLP pipeline for bias: model embeddings, model suggestions, and output reviews.

**Reasons To Reject:**

I have some concerns regarding the evaluation setup. Given that each group contains different sets of users, it seems that this can additionally affect results when comparing LLM usage across groups. This concern also stems from Table 1, which shows that there is a large variation in age and education across the groups.

My understanding is that users are evaluating each others’ business models in G0 and a pre-defined business model in G1-G4. This additional variation across groups can additionally influence results when compared to G0.

For me, it would make more sense to evaluate biases in the students’ reviews, then provide students with the interface to edit their reviews with the help of a LLM and evaluate their edited reviews for bias against their original reviews.

It would be nice to show some concrete examples of downstream gender bias that can occur due to the usage of LLMs.

**Reproducibility:**

2: Would be hard pressed to reproduce the results. The contribution depends on data that are simply not available outside the author's institution or consortium; not enough details are provided.

**Reviewer Confidence:**

4: Quite sure. I tried to check the important points carefully. It's unlikely, though conceivable, that I missed something that should affect my ratings.

**Typos Grammar Style And Presentation Improvements:**

Line 410: For “an” infinite context?

Line 513: finetuned

---

> ### Author Rebuttal · Authors · 2023-08-28
>
> Dear Reviewer 3,
>
> We thank you for the constructive feedback and the valuable time you spent reviewing this paper. We appreciate and agree with your perspective on the strengths of the paper: evaluating bias across multiple stages of the LLM pipeline, using multiple models and metrics, and highlighting a relevant and increasingly important use case of LLMs in society. We address your suggestions point-by-point below and contribute several **new** analyses to bolster the paper:
> 1) We have run new sentence-level bias analyses for 6 aligned SEAT tests, further confirming our findings of no significant bias transfer.
> 2) We [publicly provide](https://anonymous.4open.science/r/Gender-bias-in-LLMs-AD7D/German%20SEAT/sent-weat1.jsonl) the German translation of SEAT tests, revised by two native German speakers.
> 3) We [publicly provide](https://anonymous.4open.science/r/Gender-bias-in-LLMs-AD7D/README.md) the business model case study, the GPT suggestions for each group, and the cleaned and anonymized student reviews for groups G1 through G4. This dataset is provided in addition to the code for cleaning and analysis. We believe this will be a valuable resource and dataset for the NLP research community.
> 4) We conduct statistical tests (Chi-Squared for the categorical variables gender and education level, Kruskal-Wallis for age) to check for significant differences between demographic attributes (gender, age, education level) of groups G1-G4, finding no statistically significant difference between the populations.
>
> We believe that these new resources (dataset, code, study materials, sentence-level bias analysis, German translations for SEAT tests, and statistical analyses regarding study populations) increase our contribution to the field.
>
>
> **Concerning the reviewer's suggestion to first evaluate biases in the students' original reviews and then compare them with the LLM-assisted reviews**, we appreciate the interesting alternative methodology. However, our study aims to explore the influence of LLMs on students' writing in a more realistic setting. Peer-reviewing scenarios conducted in MOOCs often have students receive real-time assistance during the initial writing process rather than after the fact (which is more of an autograding task). We intend to examine LLMs in their role as an interactive tutor and writing collaborator as opposed to a grader. Consequently, the suggested framework is designed to predominantly assess students' reflective capacities rather than their interactions with assistance systems. Our approach captures how biases may or may not transfer when students use LLMs in real time, thereby offering more generalizable and ecologically valid insights.
>
>
> In order to control for the content aspect (as the content of the provided business model could inadvertently have an influence on the expected bias), we present each participant exactly **the same predefined business model**. We therefore conducted a randomized control study on Prolific: participants were divided into four groups (G1-G4), controlling for the sensitive variables of education level, language, age, and gender. While there might be (small) observable differences in the sample means or medians for these variables, additional statistical analysis confirms that there are no significant differences between groups and the randomization has worked correctly. For the age attribute, a Shapiro-Wilk test indicates that the data is not normally distributed and we have therefore conducted a non-parametric Kruskal-Wallis test, which confirms that there are no significant differences between groups G1-G4 in terms of age (H = 2.24, p = .52). Furthermore, a Chi-Square test confirms that there are also no significant differences between the four Prolific groups in terms of gender (X2 = 0.0149, p = .99) or education level (X2 = 6.65, p = .35).
> We acknowledge that the setting and population of G0 are different from the settings and populations of G1-G4, and make this explicitly clear in our analysis. We did not find it ethical to adversely impact real-world students (G0) in their core curricula in the case the bias did transfer from LLMs to students. Therefore, we aimed to recruit a comparable student population on Prolific for examining downstream bias transfer from LLMs. **Notably, in our analyses, we do not directly compare G0 results to the rest of the G1-G4 results, and in our figures, we include clear separations of G0 from the Prolific study.**
>
> **Concerning the different contexts in which the business models are being reviewed** (peer reviews in G0 vs. predefined models in G1-G4), we understand your concern. However, it's essential to clarify that despite the difference in the source of the peer reviews, both formats were designed to be rigorously equivalent in terms of complexity, content, and length. The aim of this design was to isolate the effect of the language model's suggestions on writing, thereby keeping the task as consistent as possible across all groups in the Prolific study. Our work aims to use G1-G4 to build confidence with AI-assisted writing tools so that they can be used in the real classroom in a future study without adversely affecting the students in G0.
>
> **Concerning the question of whether each user was shown the same predefined business model for the Prolific studies**, we appreciate that the reviewer points out an aspect that could be considered a variable in the experiment. In the study, all participants in the Prolific groups (G1–G4) were **exposed to the exact same predefined business model** on ski instruction to maintain task uniformity and comparability across groups. We aimed to control for the content variable so that any observed effects or lack thereof could be attributed more directly to the writing assistance provided by the different LLMs (as opposed to the content of the business model itself). This methodological choice is crucial for isolating the impact of writing assistance on potential bias transfer. We will clarify this aspect in the revised manuscript to eliminate any ambiguity.
>
> **Concerning the question about the domain-independence of the business models under review**, we appreciate the reviewer's scrutiny. We have now included the materials from the ski business case study in our repository to be released alongside the paper. The business model chosen for the study was taken from a real-world classroom **deliberately chosen to be as neutral as possible in relation to gender-biased topics**. The objective was to minimize external variables that might contribute to gender bias in the reviews. Nonetheless, your query highlights a valuable point, and we will include more examples of business model topics (technology, healthcare, food services) in the revised paper to further discuss the nuances of the German word genders in these cases as an extension of the discussion about this raised directly in the Limitations section. This will strengthen the validity of our claim that any detected bias is largely a result of the LLMs or other elements in the writing support pipeline, rather than the subjects under review. Additionally, it is imperative to note that given that all experimental groups were exposed to an identical business model, any potential impact this might have would be uniformly distributed across all groups. Given the complete randomization of our experiment, the sole distinction between groups lies in the treatment they received. Consequently, by definition, any observed differences between groups can be causally attributed to the treatment.
>
> **Concerning the reviewer's request for examples of downstream gender bias that occurs due to the usage of LLMs**, we appreciate the suggestion for enriching the content in a broader perspective. We address the reviewer’s concern with a discussion of multiple examples where downstream gender bias can be measured within the educational context. The focus of our paper is to empirically investigate whether bias is transferred from LLMs to human writing in **educational** settings. We explicitly designed a study to build confidence for whether LLMs can be used without adverse gender bias effects in the classroom instead of a less supported statement on a broader context of downstream tasks. There has been a large rise in AI in education research, specifically focusing on language models, in the last 3 years [3, 4]. To bridge the gap from research to practice, studies focusing on the safety of these models in this exact applied context of education are necessary for the field to move forward, especially as the participants of educational technology tools are often underage.
>
> Within the context of educational tasks, there are many ways to measure gender bias in classroom writing. The most prevalent way is to measure real-time assistance on writing samples (i.e. peer reviews), but other strategies include measuring changing bias across a time series of reflective journaling, analyzing emotion in essay-writing on gender-sensitive topics, discussion forum interactions with classmates from different demographics, and psychological word association tests. We will include this larger discussion of concrete examples of downstream educational tasks to measure bias.
>
> [1] Philip J. Guo and Katharina Reinecke. 2014. Demographic differences in how students navigate through MOOCs. In Proceedings of the first ACM conference on Learning @ scale conference (L@S '14). Association for Computing Machinery, New York, NY, USA, 21–30. https://doi.org/10.1145/2556325.2566247
>
> [2] Tomlinson, Carol Ann. "Teaching for excellence in academically diverse classrooms." Society 52.3 (2015): 203-209.
>
> [3] Zhai, Xuesong, Xiaoyan Chu, Ching Sing Chai, Morris Siu Yung Jong, Andreja Istenic, Michael Spector, Jia-Bao Liu, Jing Yuan, and Yan Li. "A Review of Artificial Intelligence (AI) in Education from 2010 to 2020." Complexity 2021 (2021): 1-18.
>
> [4] Lo, C. K. (2023). What is the impact of ChatGPT on education? A rapid review of the literature. Education Sciences, 13(4), 410.

---

### Official Review · Reviewer_TWkC · 2023-08-05

**Soundness:** 3

**Excitement:**

3: Ambivalent: It has merits (e.g., it reports state-of-the-art results, the idea is nice), but there are key weaknesses (e.g., it describes incremental work), and it can significantly benefit from another round of revision. However, I won't object to accepting it if my co-reviewers champion it.

**Paper Topic And Main Contributions:**

This paper investigates whether bias in LLM affects human writing. They conduct a user study with 231 students, which are divided into five groups: one with suggestions from recommender systems, one with no assistance, two with suggestions from GPT2 and GPT3 fine-tuned on non-biased corpus, and one with suggestions from pre-trained GPT3.5. By analyzing the reviews, suggestions, and model embeddings, they claim that there is no significant difference in gender bias between the peer reviews of groups with and without LLM suggestions.

**Reasons To Accept:**

1. It is interesting to explore the impact of bias from LLMs on student writing.
2. The paper is clear and well-structured.

**Reasons To Reject:**

To evaluate the bias, maybe it would be better to use sentence-level bias metrics, e.g. SEAT rather than word-level metrics.

**Reproducibility:**

3: Could reproduce the results with some difficulty. The settings of parameters are underspecified or subjectively determined; the training/evaluation data are not widely available.

**Reviewer Confidence:**

3: Pretty sure, but there's a chance I missed something. Although I have a good feel for this area in general, I did not carefully check the paper's details, e.g., the math, experimental design, or novelty.

---

> ### Author Rebuttal · Authors · 2023-08-28
>
> Dear Reviewer 2,
>
> We thank you for the positive view of the work's motivation and writing structure, the constructive feedback, and the valuable time you spent reviewing this paper. We address your suggestions point-by-point below and contribute several **new** analyses to bolster the paper:
> 1) We have run new sentence-level bias analyses for 6 aligned SEAT tests, further confirming our findings of no significant bias transfer.
> 2) We [publicly provide](https://anonymous.4open.science/r/Gender-bias-in-LLMs-AD7D/German%20SEAT/sent-weat1.jsonl) the German translation of SEAT tests, revised by two native German speakers.
> 3) We [publicly provide](https://anonymous.4open.science/r/Gender-bias-in-LLMs-AD7D/README.md) the business model case study, the GPT suggestions for each group, and the cleaned and anonymized student reviews for groups G1 through G4. This dataset is provided in addition to the code for cleaning and analysis. We believe this will be a valuable resource and dataset for the NLP research community.
> 4) We conduct statistical tests (Chi-Squared for the categorical variables gender and education level, Kruskal-Wallis for age) to check for significant differences between demographic attributes (gender, age, education level) of groups G1-G4, finding no statistically significant difference between the populations.
>
> We believe that these new resources (dataset, code, study materials, sentence-level bias analysis, German translations for SEAT tests, and statistical analyses regarding study populations) increase our contribution to the field.
>
> **Concerning the suggestion to use sentence-level bias metrics such as SEAT rather than word-level metrics like WEAT and GenBit**, we appreciate the input for a more comprehensive evaluation. We have taken this into account, and have now added new results from running a SEAT analysis on our model using the implementation from [Fairpy](https://github.com/hrishikeshvish/fairpy) (and plan to add it directly into the paper in Section 4.2). To understand why we originally did not include sentence-level bias metrics, we emphasize that the focus of our study was to investigate how biases from LLMs transfer into real-world writing tasks, which are fundamentally compositional in nature. Word-level metrics were specifically chosen because they allow for a nuanced understanding of how individual lexical choices—suggested by the LLMs—might or might not contribute to an overall biased narrative in students' writing. Word-level metrics offer granularity, allowing us to pinpoint exact instances of bias within the language model embeddings, a level of detail that is often lost in sentence-level evaluations. Moreover, WEAT and GenBit have been more widely validated than any sentence bias measures and are capable of identifying subtle biases in embeddings. Using sentence-level metrics could potentially dilute these subtle biases, offering a less granular view, especially in the context of peer reviews, which are often short and focus more on objective evaluations than narrative discourse.
>
> Despite these concerns, we believe both types of analyses have merit, and together, they provide a richer picture of bias in embeddings. **We present the results of our SEAT analysis on the fine-tuned GPT model used for G2 below**, the analogous counterpart to our WEAT analysis included in Section 4.2. SEAT is a sentence-based embedding test, computing the sentence embeddings for two sets of target sentences (in test 6, male and female names) and two sets of attribute sentences (career and family). The pairwise embedding distance between each target sentence and attribute sentence (e.g. “Das ist Hans” and “Dies ist eine Exekutive”) is computed. Across tests 6, 6b, 7, 7b, 8, and 8b, which correspond to the analogous gender tests of WEAT, there is no significant difference found between the embeddings of the target sentences and attribute sentences (i.e. in SEAT 6, male and female names are both equally similar to career and family words), computed using p-values and a hypothesis test as per the standard implementation. Each test also has an effect size, ranging from -2 to +2, with 0 representing a completely neutral effect between both target words and attribute words. Most effect sizes are within 0.1, so there are only minimal bias effects. The comparatively largest effect (SEAT test 7) is -0.705 from comparing Math and Arts in association with Male and Female terms, but this was still found to have insignificant bias. These results are in line with the WEAT analysis on the word level granularity, which also found no significant gender bias in the model embeddings.
>
> |Test|Hypothesis Test|Effect size|
> |:----|:----|:----|
> |sent-weat6|no difference between MännlicheNamen and WeiblicheNamen in association to attributes Karriere and Familie;|0.021|
> | |(no difference between Male Names and Female Names in association to attributes Career and Family)| |
> |sent-weat6b|no difference between Wissenschaft and Weibliche Begriffe in association to attributes Karriere and Familie;|-0.074|
> | |(no difference between Male Terms and Female Terms in association to attributes Career and Family)| |
> |sent-weat7|no difference between Mathe and Kunst in association to attributes Männliche and Weibliche Begriffe;|-0.705|
> | |(no difference between Math and Arts in association to attributes Male Terms and Female Terms)| |
> |sent-weat7b|no difference between Mathe and Kunst in association to attributes MännlicheNamen and WeiblicheNamen;|-0.209|
> | |(no difference between Math and Arts in association to attributes Male Names and Female Names)| |
> |sent-weat8|no difference between Wissenschaft and Kunst in association to attributes Männliche and Weibliche Begriffe;|-0.069|
> | |(no difference between Science and Arts in association to attributes Male Terms and Female Terms)| |
> |sent-weat8b|no difference between Wissenschaft and Kunst in association to attributes MännlicheNamen and WeiblicheNamen;|0.078|
> | |(no difference between Science and Arts in association to attributes Male Names and Female Names)| |

---

### Official Review · Reviewer_bbvs · 2023-08-05

**Typos Grammar Style And Presentation Improvements:** None
**Soundness:** 3

**Excitement:**

3: Ambivalent: It has merits (e.g., it reports state-of-the-art results, the idea is nice), but there are key weaknesses (e.g., it describes incremental work), and it can significantly benefit from another round of revision. However, I won't object to accepting it if my co-reviewers champion it.

**Missing References:**

None

**Paper Topic And Main Contributions:**

This paper investigated an interesting problem, i.e., how bias might transfer through large languages models in the setting of AI-human collaborative writing. To investigate this problem, the authors conducted a large-scale user study to evaluate student writing in providing peer reviews for business cases in German. Then, the bias was measured by using GenBit gender bias analysis and Word Embedding Association Tests. The key contribution of this study is that, to my knowledge, this is the first study investigating the potential bias that might seep into educational tasks and affect students' writing styles and perspectives. The study is well designed and the presented findings are somewhat meaningful.

**Questions For The Authors:**

None

**Reasons To Accept:**

1. To my knowledge, this is the first study investigating the potential bias that might seep into educational tasks and affect students' writing styles and perspectives.
2. The study was well designed and implemented.

**Reasons To Reject:**

(A) As indicated by the authors, a key limitation is that this study was based on one single writing task, which inherently limits the contributions of the paper.

**Reproducibility:**

3: Could reproduce the results with some difficulty. The settings of parameters are underspecified or subjectively determined; the training/evaluation data are not widely available.

**Reviewer Confidence:**

4: Quite sure. I tried to check the important points carefully. It's unlikely, though conceivable, that I missed something that should affect my ratings.

---

> ### Author Rebuttal · Authors · 2023-08-28
>
> Dear Reviewer 1,
>
> We thank you for the positive sentiment regarding the study design and motivation, interesting questions, and the valuable time you spent reviewing this paper. We address your suggestions point-by-point below and contribute several **new** analyses to bolster the paper:
> 1) We have run new sentence-level bias analyses for 6 aligned SEAT tests, further confirming our findings of no significant bias transfer.
> 2) We [publicly provide](https://anonymous.4open.science/r/Gender-bias-in-LLMs-AD7D/German%20SEAT/sent-weat1.jsonl) the German translation of SEAT tests, revised by two native German speakers.
> 3) We [publicly provide](https://anonymous.4open.science/r/Gender-bias-in-LLMs-AD7D/README.md) the business model case study, the GPT suggestions for each group, and the cleaned and anonymized student reviews for groups G1 through G4. This dataset is provided in addition to the code for cleaning and analysis. We believe this will be a valuable resource and dataset for the NLP research community.
> 4) We conduct statistical tests (Chi-Squared for the categorical variables gender and education level, Kruskal-Wallis for age) to check for significant differences between demographic attributes (gender, age, education level) of groups G1-G4, finding no statistically significant difference between the populations.
>
> We believe that these new resources (dataset, code, study materials, sentence-level bias analysis, German translations for SEAT tests, and statistical analyses regarding study populations) increase our contribution to the field.
>
> **Concerning the limitation that our study was based on one single writing task**: we agree with the reviewer that our experiments are limited to a single scenario. However, this narrow scope was intended as it allowed for an in-depth, controlled analysis with a larger population of students, eliminating confounding variables and enhancing the study's rigor. Our primary purpose was to have high internal validity in this first experiment in order to examine whether the study design, conduct, and analysis answer the research question. The chosen peer review task is highly representative and educationally very relevant. A recent meta-analysis showed that peer feedback has a significant effect on academic performance [1], which holds across primary, secondary as well and tertiary education. The effect of peer feedback on academic performance was shown to be significantly higher compared to a condition without feedback (g = 0.31, p = 0.004) as well as to a condition in which the teacher provided feedback (g = 0.28, p = 0.007). Other meta-analyses have further confirmed the substantive benefits of peer feedback across a wide range of contexts [2, 3], demonstrating that it can yield greater learning benefits than teacher feedback.
>
>
> The resulting data allowed us to observe the interaction between LLMs and human writing in a real-world context, thereby providing valuable insights into whether bias from LLMs transfers to human writing. We argue that even though the study focuses on one type of task, the implications of our findings are broad and pave the way for future research on the impact of AI in education across modeling, writing, and grading tasks. Our methodology lays the groundwork for future, more expansive studies, and a pipeline to study bias in educational tasks.
>
>
> [1] Double, K.S., McGrane, J.A. & Hopfenbeck, T.N. The Impact of Peer Assessment on Academic Performance: A Meta-analysis of Control Group Studies. Educ Psychol Rev 32, 481–509 (2020). https://doi.org/10.1007/s10648-019-09510-3
>
> [2] Yan, Zi, et al. "Effects of self-assessment and peer-assessment interventions on academic performance: A pairwise and network meta-analysis." Educational Research Review (2022): 100484.
>
> [3] Li, Hongli, et al. "Does peer assessment promote student learning? A meta-analysis." Assessment & Evaluation in Higher Education 45.2 (2020): 193-211.

---

### Meta-Review · Area_Chair_ab4N · 2023-09-19

**Recommendation:** 3

**Metareview:**

The reviewers found the problem interesting and the subject of the paper novel. The methodology was seen as fairly comprehensive and sound overall, and the paper’s writing is clear and easy to follow. There were some concerns about the evaluation setup, but these appear to be addressed by the authors in their rebuttals, so I believe the soundness should not be negatively affected by most of these evaluation issues. One reviewer still felt that the scope of the paper was a bit limited as the paper focused on a single task. None of the reviewers were particularly excited about the paper (all felt ambivalent).

---

### Decision · Program_Chairs · 2023-10-07

**Decision:**

Accept-Findings

**Comment:**

The reviewers found the problem interesting and the subject of the paper novel. The methodology was seen as fairly comprehensive and sound overall, and the paper’s writing is clear and easy to follow. There were some concerns about the evaluation setup, but these appear to be addressed by the authors in their rebuttals, so I believe the soundness should not be negatively affected by most of these evaluation issues. One reviewer still felt that the scope of the paper was a bit limited as the paper focused on a single task. None of the reviewers were particularly excited about the paper (all felt ambivalent).